# Direct observation of the on-site oxygen $2p$ two-hole Coulomb energy in La$_2$CuO$_4$

Danilo Kühn [1,2] ✉, Swarnshikha Sinha[1,2,3], Fredrik O. L. Johansson [2,4], Katarzyna Siewierska[1], Antonello Tebano[5,6], Nils Mårtensson[2,4], Andreas Lindblad [2,4], Daniele Di Castro[5,6] & Alexander Föhlisch [1,2,3] ✉

Electron correlation in functional materials has remained a challenge with strong deviations of electronic structure from mean field approaches. In high temperature superconductors the electron-electron and hole-hole interaction energies are essential in the underlying pairing mechanisms. For cuprates, oxygen holes have been considered of central importance for superconductivity. In La$_2$CuO$_4$ the site specific oxygen $2p$ hole-hole Coulomb energy has been determined by Auger photoelectron coincidence spectroscopy. This experimental approach allows to separate the different oxygen sites, i.e. the lattice oxygen, and distinguish from otherwise overlapping signal from surface oxygen. Values of 6.3 ± 0.2 eV for oxygen in the Cu-O planes and an upper limit of 9.2 ± 0.2 eV for apical oxygen are found to be on the high energy side of reported computational values and narrows the range of experimentally reported values. Additionally, a much reduced hybridization in La$_2$CuO$_4$ as compared to CuO is found in O $2p$ hybridization strengths.

Electron correlation in functional materials has remained a challenge from a computational and experimental standpoint. This interaction between electrons brings strong deviations in terms of electronic structure from established mean field approaches[1–4]. In high temperature superconductors the electron-electron and hole-hole interaction energies are thought to be essential in the underlying pairing mechanisms[5,6]. In the family of cuprates the oxygen holes are important for superconductivity[7–10]. Experimentally, single particle transitions and detection have been largely insensitive to trace multielectron interactions and correlation.

Layered copper oxides, among other transition metal oxides (TMO's), have exceptional properties and phase diagrams involving superconductivity, anti-ferromagnetic order, strange-metal and pseudo-gap phases as well as charge-density waves that standard band theory cannot explain. Instead, correlations within and between the electronic, lattice and magnetic subsystems are essential[11]. Thus, a variety of models have been developed, i.e. one-band[12] and three-band[13,14] Hubbard models, the t-J model[15], the resonating valence bond model[16], the marginal Fermi liquid theory[17] and pair-density waves[10,18] among others. However, no definitive consent about the exact pairing mechanism for the Cooper pair formation in the superconducting state has been reached. Although the phonon contribution to the physics of high-temperature superconductors (HTSC) cannot be entirely neglected[19,20], it is becoming evident that spin fluctuations are one of the main ingredients of the pairing mechanism in cuprates[21], as suggested by the correlation found between T$_c$ and the exchange interaction J in some cuprate families[22–25] and by the finding that short range spin correlations survive well within the superconducting phase[26,27]. The connection between J and the parameters relevant for high energy electronic excitations, which govern the electronic properties of the parental compounds of HTSC's, remains an important open issue[28,29].

The Zaanen-Sawatzky-Allen (ZSA) classification[1] has served as a powerful conceptional tool to characterize the electronic structure

[1]Institut für Methoden und Instrumentierung der Forschung mit Synchrotronstrahlung, Helmholtz-Zentrum Berlin für Materialien und Energie GmbH, Berlin, Germany. [2]Uppsala-Berlin Joint Laboratory on Next Generation Photoelectron Spectroscopy, Berlin, Germany. [3]Institut für Physik und Astronomie, Universität Potsdam, Potsdam, Germany. [4]Department of Physics and Astronomy, Division of X-ray Photon Science, Uppsala University, Uppsala, Sweden. [5]Dipartimento di Ingegneria Civile e Ingegneria Informatica, Universitá di Roma Tor Vergata, Roma, Italy. [6]CNR-SPIN, Universitá di Roma Tor Vergata, Roma, Italia. ✉e-mail: danilo.kuehn@helmholtz-berlin.de; alexander.foehlisch@helmholtz-berlin.de

of TMO's. A high Coulomb repulsion energy between the transition metal d-electrons $U_{dd}$ can split the narrow metal d band into two sub bands which hybridize with the O 2p band. The relation between $U_{dd}$ and $\Delta$, the energy required to transfer an electron from the ligand to the metal, as well as the hybridization interaction strength and the ligand band width, lead to either a metallic conductor, a Mott-insulator, a charge transfer insulator or a semi-metal. The electronic parameters $U_{dd}$ and $\Delta$ have been investigated for a variety of transition metal oxides and cuprate superconductors by (inverse) photoelectron spectroscopy (IPES/ PES) and Auger electron spectroscopy (AES)[30–32].

Although electron correlation in the oxygen valence band is often neglected due to the large extension of the O 2p orbitals, some researchers reported a significant on-site Coulomb repulsion energy $U_{pp}$, potentially relevant for the correct description of the electronic ground state of HTSC's[30,33]. Furthermore, on-site O $2p^4$ two-hole valence states should be highly sensitive to the covalency of the involved orbitals[30,34,35]. Quantitative determination of $U_{pp}$ has remained experimentally challenging, since classical electronic structure tools, such as valence band photoemission, trace the single-hole final state with no access to the on-site two-hole final state and Coulomb repulsion.

In this work, we use Auger photoelectron coincidence spectroscopy (APECS) for the direct experimental determination of the on-site lattice oxygen 2p two-hole states in La$_2$CuO$_4$ (LCO) in direct comparison to CuO. The unprecedented chemical selectivity allows to separate the surface oxygen contributions from the lattice oxygen contributions fully on both crystals. We then determine the respective on-site Coulomb repulsion energies $U_{pp}$ of lattice oxygen and discuss them in the context of electronic structure calculations. Furthermore, we observe a high itinerance of the two O 2p holes in CuO, reflecting its 3-dimensional electronic structure. In contrast, the two O 2p holes are predominantly localized in LCO, probably as a consequence of the ionic character of the two-dimensional electronic structure of LCO. This results in a high ratio of $U_{pp}$ to band width, that is put into perspective within the established Cini-Sawatzky model and literature.[36–38] Our findings are relevant for advanced electronic structure models to understand the role of electron correlation for the pairing in cuprate high temperature superconductors and a variety of strongly correlated oxide compounds.

## Results and discussion

Copper(II) oxide (CuO) crystallizes in a monoclinic lattice and has an antiferromagnetic ground state with a band gap of about 1.4 eV despite the incomplete $3d^9$ shell filling. This is explained by strong electron correlations, which localize the charge carriers in the d band. In the ZSA model, CuO is classified as a charge transfer insulator with a high $U_{dd} = 8 − 9$ eV, which exceeds $\Delta = 2.2$ eV[1,32]. LCO consists of stacks of LaO · CuO$_2$ · LaO planes, where each Cu atom is octahedrally coordinated to six oxygen atoms[39]. The oxygen and copper atoms are formally in a divalent ionic state, however with strong hybridization in the CuO$_2$ planes, whereas the apical oxygen in the La-O planes is weakly hybridized[40,41]. In the undoped phase, the ground state has antiferromagnetic ordering with the magnetic moments localized in the $3d^9$ shell. Upon p-doping, holes in the oxygen p shell provide free charge carriers enabling high Tc superconductivity up to about 40 K at optimum doping levels.

Figure 1a shows how the on-site oxygen 2p two-hole states are prepared and detected with APECS. In a simplified two-step picture the O 2p double photoionization process can be rationalized as the oxygen KVV Auger decay following oxygen 1s core-ionization. Since the emitted electron pairs are detected in coincidence, they truly contain processes that are leading to the on-site oxygen 2p two-hole final state thus eliminating overlapping features and background from non-local contributions that are present in classical Auger electron spectroscopy[42–44]. Technically, the required coincidence detection efficiency is gained by the bespoke Coincidence ESCA experiment at the UE52-PGM beamline at the BESSY II electron storage ring[45] with two Angle Resolved Time of Flight (ARTOF) Electron Analysers. The surface termination of TMO's and HTSC's gives rise to chemically inequivalent species of lattice oxygen atoms O$_{latt.}$ and under-coordinated surface oxygen atoms (O$_{ad.}$), as shown in Fig. 1b. The approach of APECS not only prepares the on-site oxygen 2p two-hole state by coincidences of oxygen 1s (K-shell) ionization and the oxygen KVV Auger decay, but also allows to separate within the coincidence map the contributions from lattice oxygen O$_{latt.}$ and surface oxygen O$_{ad.}$ species completely. This is a particular advantage for the investigation of oxide surfaces because of the surface termination and the fact that in a variety of TMO's and cuprates it is difficult to prepare flat surfaces without step edges and free of defect states[30,46,47].

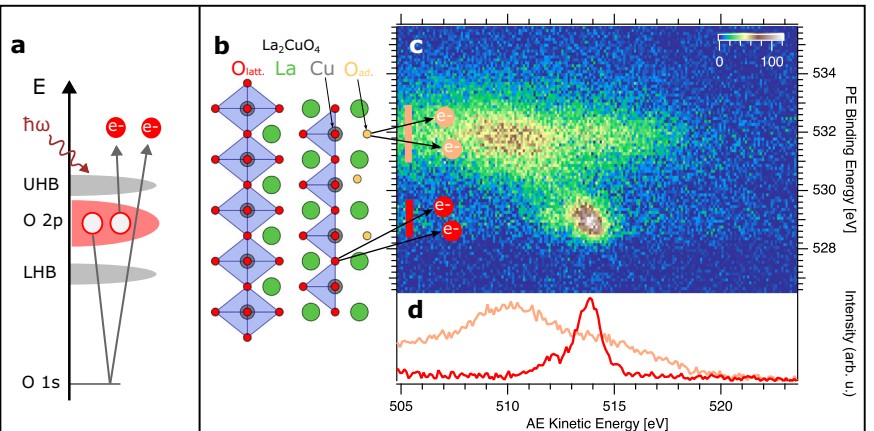

**Fig. 1 | Separation of surface and lattice oxygen through Auger photoelectron coincidence spectroscopy (APECS). a** The on-site oxygen 2p two-hole state is prepared and detected via coincidences of oxygen 1s (K-shell) ionization and the oxygen KVV Auger decay. LHB and UHB denote lower and upper Hubbard band, respectively. **b** Structure and surface termination of tetragonal La$_2$CuO$_4$: Octahedral coordination (blue polygons) of Copper atoms (grey) to six lattice oxygen atoms O$_{latt.}$ (red). The surface layer (right hand side) contains under-coordinated surface oxygen species O$_{ad.}$ (light orange). **c** Oxygen 1s/ KVV coincidence map of La$_2$CuO$_4$ with distinctly separated lattice oxygen O$_{latt.}$ and surface oxygen O$_{ad.}$ on-site oxygen 2p two-hole state spectral contributions. Color bar shows counts. **d** Coincidence Auger electron spectra of O$_{latt.}$ (red) and O$_{ad.}$ (light orange) obtained by integrating the map in **c** over the respective photoelectron binding energy ranges marked with vertical bars in the same color. Source data are provided as a Source Data file.

Figure 1 c shows the two dimensional coincidence map of O 1s photoelectron and O KVV Auger electron pairs emitted from $O_{latt.}$ and $O_{ad.}$ species of the $La_2CuO_4$ sample. Photoelectrons are shown in binding energy and Auger electrons in kinetic energy referenced to the Fermi level. A background from uncorrelated electron pairs (accidental coincidences) has been removed (see Methods for details). Electron pairs emitted from oxygen ions in the crystal lattice ($O_{latt.}$) are centred at about $E_b$ = 529 eV, in agreement with previous XPS studies[48,49], and $E_{kin}$ = 514 eV. They are fully separated from the surface oxygen $O_{ad.}$ species and from oxygen containing adsorbed molecules, which appear in a broad feature at about $E_b$ = 532 eV and $E_{kin}$ = 510 eV. The depth distribution of these features can be directly inferred from their relative intensities in the very surface sensitive coincidence measurement in comparison to the deeper probing non-coincidence measurement (see Supplementary Fig. 1).

The relatively high intensity from $O_{ad.}$ is due to the small APECS probing depth of a few Å[43,50,51]. Figure 1d shows how one by partial integration can create pure on-site two-hole spectra from lattice oxygen and surface oxygen, respectively. This separation is commonly impossible with non-coincidence Auger electron spectroscopy due to spectral overlap. The integration ranges of the photoelectron binding energies that are associated with either the lattice or the surface oxygen contributions are indicated with bars in the coincidence map in Fig. 1c.

Figure 2 shows O 1s/ KVV coincidence maps of CuO (Fig. 2a) and LCO (Fig. 2d) in direct comparison. The CuO and LCO maps have both well separated surface and bulk oxygen spectral features. In CuO lattice oxygen is centered at $E_b$ = 529.5 eV, $E_{kin}$ = 512.5 eV. Non-stoichiometric excess oxygen at the surface is found at $E_b$ = 531.2 eV, $E_{kin}$ = 510.5 eV[32]. In LCO the surface and lattice oxygen assignment is alike, as discussed already in Fig. 1c. Figure 2b and c shows Auger

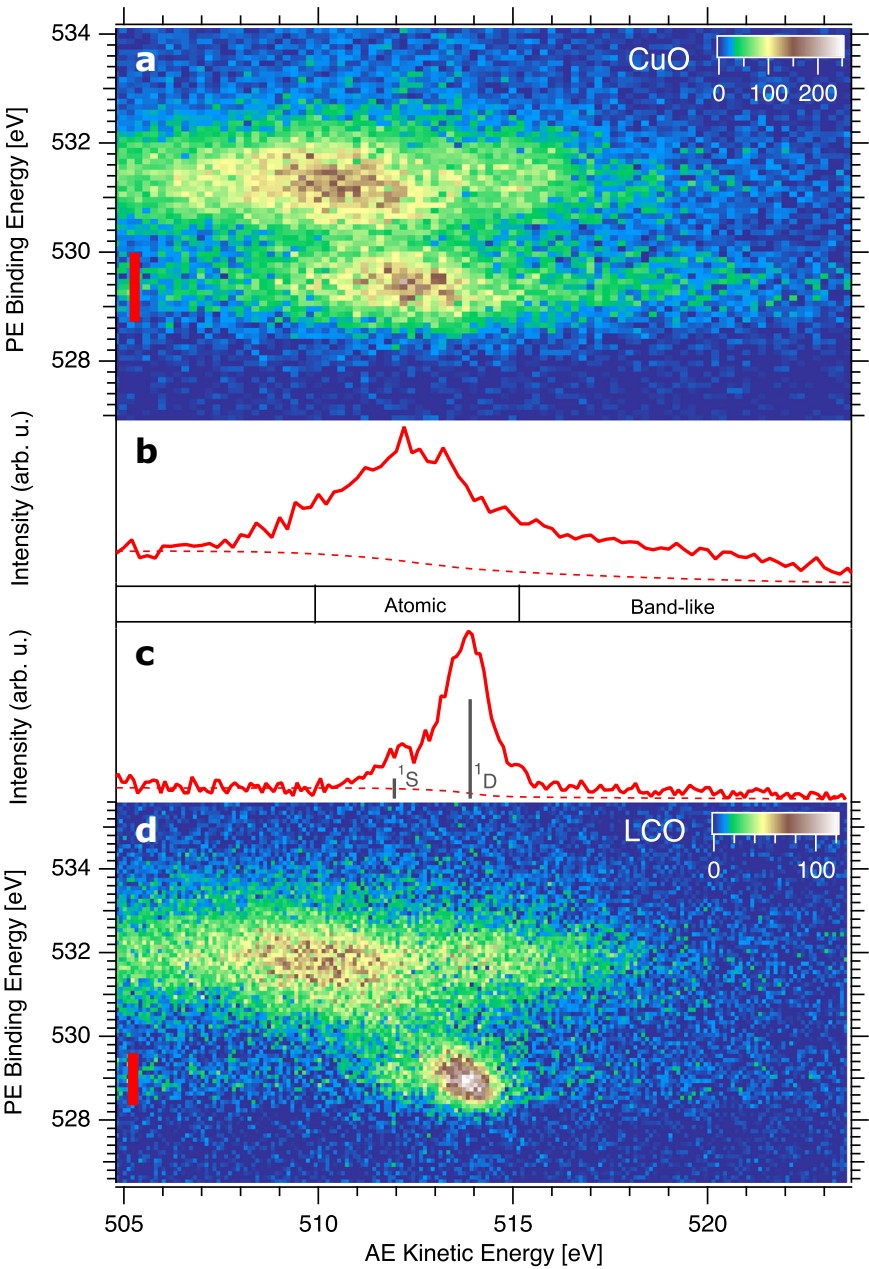

**Fig. 2 | APECS measurements of CuO and La₂CuO₄. a**, **d** O 1s/ KVV coincidence maps of CuO (**a**) and La₂CuO₄ (**d**). Color bars show counts. **b**, **c** Coincidence Auger electron spectra obtained by integrating over the PE binding energy of lattice oxygen in (**a**) and (**d**), respectively. The integration ranges are shown with red bars in (**a**) and (**d**). The dashed lines show the Shirley background of the spectra and the positions of atomic multiplet states are labeled for La₂CuO₄. Source data are provided as a Source Data file.

electron spectra derived from the 2D maps of CuO and LCO, respectively, by integrating over the photoelectron part from the lattice oxygen of each crystal (marked with red bars). The direct comparison of the lattice oxygen two-hole final states of CuO and LCO reveals striking differences. The CuO spectrum consists of a broad peak at 513 eV which contains mostly atomic localized O $2p^4$ states and due to covalent coupling to copper sites possibly some O $2p^5 3d^8$ final states. In addition, a pronounced tail to higher kinetic energy, also called 'band-like part', indicates a high probability for the two holes to delocalize within the oxygen band. Such oxygen KVV spectral shape has been reported for several three-dimensional metal oxides[31,32,52,53].

The LCO spectrum is dominated by a well resolved $^1S$ and $^1D$ atomic multiplet split by 2 eV at 512 eV and 514 eV, respectively. This is in good agreement with optical data (2.2 eV)[54]. Next to these intense atomic peaks a weak band-like part at higher kinetic energy is seen (Supplementary Fig. 2 for extended energy range of the band like part). Actually, LCO has two different lattice oxygen species: in-plane oxygen and apical oxygen. They have not been separated by conventional PES and AES[48] and even with our increased chemical sensitivity of APECS, the LCO measurements find no discernable twins of atomic ($^1D$;$^1S$) multiplets, but only one set of atomic multiplet fine structure, containing both oxygen species. This implies that the in-plane oxygen to apical oxygen two-hole final state energy difference is below the experimental resolution. Overall, the narrow atomic two-hole final states in LCO reflect the localized ionic character of the valence states in the two-dimensional electronic structure of LCO in contrast to the three-dimensional more covalent electronic structure of CuO which leads to additional non-local final states. For further analysis, a Shirley type background[55] is subtracted from the spectra, indicated by dashed lines in Fig. 2b and c, and the kinetic energy is converted to two-hole binding energy by subtracting the O 1s binding energy.

The Cini-Sawatzky (CS) model provides a powerful framework to obtain $U_{pp}$ of pure lattice oxygen from experimental spectra of two-hole states in O $2p$ orbitals[56]. Within this model, $U_{pp}$ is described by the energy difference between the bound (correlated) state and the band-like (uncorrelated) part of the O $2p$ two-hole spectrum. Cini and Sawatzky provided numerical expressions of the full spectral distribution simulating the local-interacting two-particle density of states from non-interacting one-particle DOS[36–38]. Experimentally, the latter can be the oxygen derived valence spectral states[17,32,57,58] or computationally, the oxygen partial density of states (PDOS)[48,59].

In Fig. 3, we use the Cini-Sawatzky expression (Eq. (1)) to simulate and fit the two-hole spectra in order to extract $U_{pp}$:

$$N_U(E) = \frac{N_0(E)}{[1 - U \cdot H(E)]^2 + \pi^2 U^2 N_0(E)^2} \qquad (1)$$

$N_U(E)$ is the interacting two-particle DOS, $N_0(E)$ is the self-convolution of the non-interacting single particle DOS, U is the Coulomb repulsion and $H(E)$ is the Hilbert transform of $N_0(E)$. For the single particle DOS we use the O $2p$ PDOS from band structure calculations by Pickett et al.[40] for LCO and by Ching et al.[60] for CuO (see Supplementary Fig. 4). $N_0(E)$, which would represent the two-hole spectra without electron correlation ($U_{pp} = 0$ eV), is shown in Fig. 3 for each simulation. Two CS shapes are used to model the $^1S$ and $^1D$ atomic multiplets with the respective O $2p$ PDOS. A similar approach was previously used to model APECS spectra from transition metal surfaces[61–63] and magnetic thin films[64,65] in order to probe electronic correlations.

As mentioned, the experimental O $2p$ two-hole spectra of in-plane and apical oxygen of LCO are spectroscopically overlapping and indistinguishable in our APECS measurement. Therefore, we perform two separate CS fits to the spectrum, one based on the calculated PDOS of in-plane oxygen and a second with PDOS for apical oxygen. For in-plane oxygen of LCO (see Fig. 3a), we obtain an optimum fit for $U_{pp}$ ($^1D$) = 6.3 ± 0.2 eV and $U_{pp}$ ($^1S$) = 8.6 ± 0.2 eV, an intensity ratio I($^1D$)/

I($^1S$) = 4.9 ± 0.2 and a broadening of 1.32 ± 0.02 eV. This broadening accounts for effects that are not captured by the Cini-Sawatzky model: After removing experimental contributions, the line width of the multiplet states is similar to the O 1s width (1.1 eV) of lattice oxygen, which suggests a related broadening mechanism present in O 1s and O KVV peaks. Therefore, we favor to attribute the broadening to phonon excitations or small variations of the local potential, which should both effect the PES and AES similarly[66]. Also a small life time broadening and dispersion of the two-hole state might contribute. The $U_{pp}$ error bar comprises the overall experimental uncertainty (0.1 eV) and the uncertainty in digitization of the DOS (0.1 eV), whereas the uncertainty of the fit is negligible (0.01 eV) (see Methods for details). The uncertainties of I($^1D$)/I($^1S$) and of the broadening are given by standard deviations of the fit parameters. With this fit model, the positions of the atomic multiplet and the shape and intensity of the band like part are quantitatively matching the experimental spectrum. The intensity ratio $^1D$/$^1S$ is in fair agreement with calculations of atomic Auger transition rates for Neon[67] I($^1D$)/I($^1S$) = 6.0, which has formally the same $2p^6$ valency as the lattice oxygen ions. For apical oxygen (see Fig. 3b), we obtain $U_{pp}$ ($^1D$) = 9.2 ± 0.2 eV, $U_{pp}$ ($^1S$) = 11.3 ± 0.2 eV, I($^1D$)/I($^1S$) = 4.6 ± 0.2 and a broadening of 1.50 ± 0.02 eV. The fit quality of the band-like part region is substantially reduced in the CS fit based on the PDOS of apical oxygen. From this, we conclude that the major contribution in the experimental O $2p$ two-hole spectrum arises from in-plane oxygen. This would comply with the most stable La-O (top layer), Cu-O (second layer), La-O (third layer) surface termination. The signal from the under-coordinated oxygen of the top layer will not appear in the spectrum since we discriminate against it in APECS. The intensity of in-plane oxygen from the second layer is expected to be substantially higher in the lattice oxygen O $2p$ two-hole spectrum than the intensity from apical oxygen in the third layer due to the high surface sensitivity of our APECS measurement[51].

Figure 3c shows Cini-Sawatzky fits to CuO. Here, we could not get a satisfactory fit over the full energy range with two CS components ($^1D$; $^1S$). Hence, we have used two different approaches. First the fit was restricted to the energy range of the bound state (15-22 eV). Here, we obtain $U_{pp}$ ($^1D$)=6.9 eV. Since the atomic $^1D$ and $^1S$ states can not be resolved, the difference of $U_{pp}$ ($^1S$) and $U_{pp}$ ($^1D$) is fixed to 2.2 eV obtained from the well resolved atomic multiplet of LCO and the intensity ratio $^1D$/$^1S$ is fixed to 4.9. The fitted Gaussian broadening of 4.3 eV is substantially bigger than in LCO and than the O 1s line width of CuO lattice oxygen. This suggests effects beyond phonon broadening or variation of the local potential, as, e.g., coupling to the copper ions. Also the band like intensity is obviously much underestimated in this fit. In a second approach we allow for an energy shift of $N_0(E)$, which leads to a reasonable fit over the full energy range for a shift of 3.0 eV, equivalent to a shift of the PDOS of 1.5 eV, and $U_{pp}$ ($^1D$)=3.3 eV. We relate all data to the common Fermi level of a metallic sample which still allows for surface shifts in doped gaped materials[66]. For LCO, defect states provide a higher charge carrier density than for CuO. It is worth to mention, that an equally good fit result can also be obtained with a DOS from a band structure calculation including correlation without the need to shift the SCDOS. Supplementary Fig. 3 shows a fit based on a calculated PDOS from Anisimov et al. in a LDA+U framework. Since $U_{pp}$ is directly related to the energy difference of the SCDOS to the bound (atomic) peak within the CS model, quite different values of $U_{pp}$ are obtained for the unshifted SCDOS as compared to the shifted SCDOS or the SCDOS obtained from LDA+U calculations. The $U_{pp}$ values of 6.9 eV and 3.3 eV obtained from the two different modelling approaches should be seen as upper and lower limits of the O $2p$ Coulomb repulsion energy in CuO, respectively.

Table 1 lists parameters from the fits. The $U_{pp}$ ($^1D$) of in-plane oxygen in LCO is overall in agreement with AES measurements by Rietveld et al.[48] (6.25 eV in LCO) and Bar-Deroma et al.[59] (5 eV in LSCO/LBCO). Also some AES and RAES measurements of other cuprates show similar $U_{pp}$

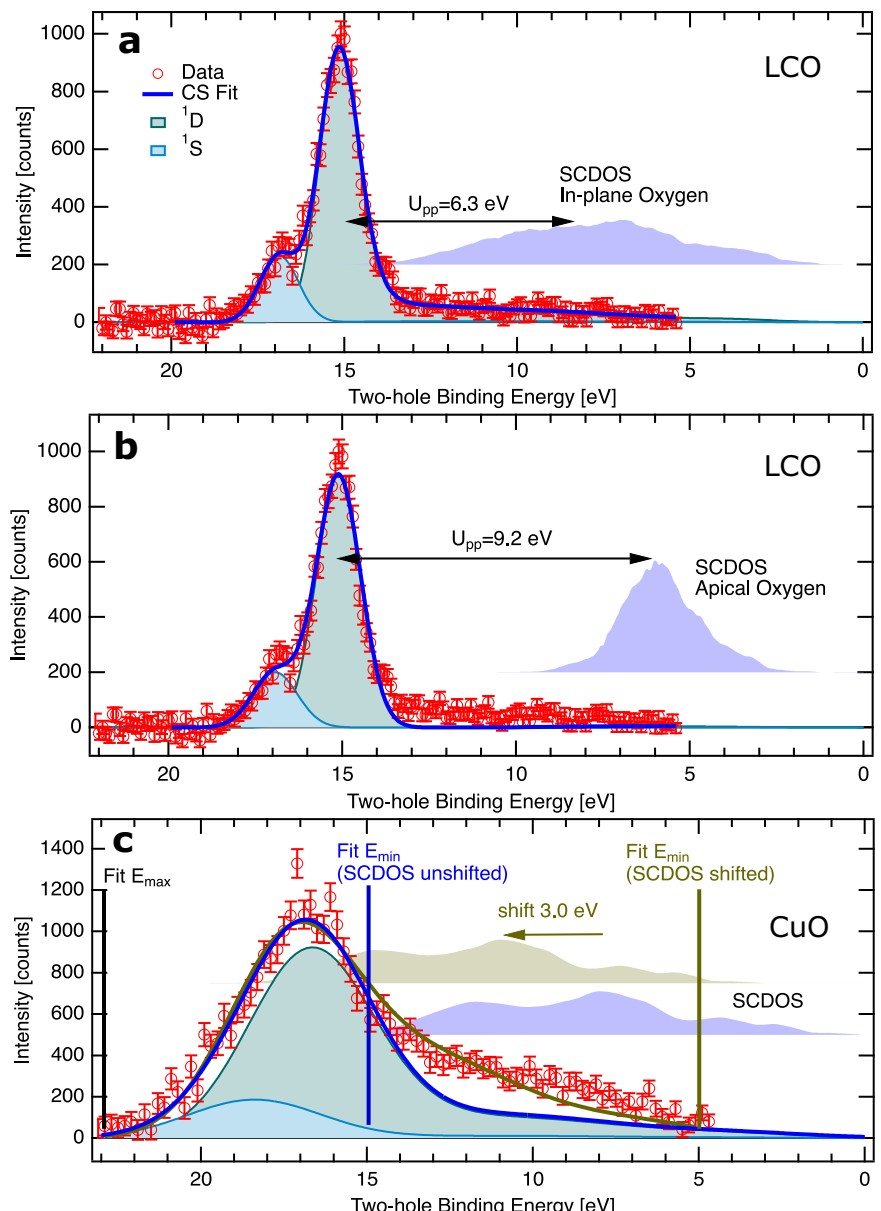

**Fig. 3 | Determination of U$_{pp}$ two-hole Coulomb repulsion in La$_2$CuO$_4$ and CuO with the Cini-Sawatzky model. a** La$_2$CuO$_4$ in-plane oxygen based on PDOS's from Pickett et al.[40]. **b** La$_2$CuO$_4$ apical oxygen based on PDOS's from Pickett et al.[40]. **c** CuO lattice oxygen with two different models (details described in the text), PDOS from Ching et al.[60]. The vertical bars in (**c**) indicate the limits of the fits. For each panel, the respective self-convoluted density of states (SCDOS's) based on the calculated PDOS's are shown vertically offset and the common legend from **a** applies. Data points with error bars are red, blue lines show fits, individual ¹S and ¹D fit components are shown with light blue and mint green shaded forms, respectively. Vertical error bars are standard deviation, see Methods section. Source data are provided as a Source Data file.

between 5 and 7 eV, i.e. YBCO[17], BSCCO[33,57,58]. Many calculations have predicted a somewhat lower $U_{pp}$ for LCO between 3.6-4.64 eV[68–70], although $U_{pp}$ = 6.1 eV found in a recent ab-initio calculation by Hirayama et al.[71] is in very good agreement with our findings. It should be noted that resonant Photo/Auger electron spectroscopy (RPES/RAES) were also used to investigate two-hole states, which appear as satellites in the valence spectrum, e.g. BSCCO[57,58,72]. Here selectivity can arise from resonant population of unoccupied lattice oxygen states and contributions from defective oxygen may be reduced. However, an accurate extraction of the two-hole states, in particular of the band-like part, can be difficult because of the overlap with the one-hole part of the valence spectrum. Furthermore, the final state in RAES contains of one additional electron in the valence, from the core-excitation process, which interacts with the two holes.

Our experimental determination of on-site Coulomb energies in O 2$p$ orbitals of LCO and CuO with Cini-Sawatzky modelling have important implications for correlated two particle states in cuprates: The on-site O 2$p$ Coulomb energy is significantly reduced in the Cu-O planes (6.3 eV) as compared to the La-O planes (9.2 eV) probably due to enhanced screening in the Cu-O planes. This might confine correlated two-hole states to the Cu-O planes in the LCO material due to the favorable energetics. Furthermore, the two-hole states in LCO are strongly localized as revealed by the strong quenching of delocalized (band-like) final states in the spectra in comparison to three-dimensional CuO. These anisotropic two-particle properties in LCO differ substantially from single particle properties, where oxygen 2$p$ PDOS DFT calculations show similar hybridization strength of oxygen in the Cu-O planes of layered LCO and of oxygen in three-dimensional

**Table 1 | Parameters of Cini-Sawatzky simulations**

|  | $U_{pp}(^1D)$ [eV] | $U_{pp}(^1S)$ [eV] | $I(^1D)/I(^1S)$ | $w_{Gauss}$ [eV] |
|---|---|---|---|---|
| LCO in-plane | 6.3 | 8.6 | 4.9 | 1.3 |
| LCO apical | 9.2 | 11.3 | 4.6 | 1.5 |
| CuO | 6.8 | 9.0 | 4.9 | 4.3 |
| CuO (shift) | 3.3 | 5.5 | 4.9 | 3.8 |

The energy shift of the SCDOS in "CuO (shift)" is 3.0 eV. Details are discussed in the main text.

CuO. We thus think that the observed properties of the Cu-O planes, with regard to two hole states, relate to the preference of pairing in cuprate high temperature superconductors in the Cu-O planes alike. A quantitative knowledge of Coulomb energies of two-hole states at oxygen sites is thus relevant for theoretical models, e.g. to link electronic parameters with superexchange[57].

We conclude with a brief discussion on the precision of $U_{pp}$ obtained from the CS fits of the experimental O $2p$ two-hole spectra. The choice of U in a CS fit, for a given SCDOS, has two major implications: First, the energy position of the atomic part is approximately determined by the position of the SCDOS maximum plus $U_{pp}$. Secondly, the intensity ratio of the bound to band like part scales with U/W. If only $U_{pp}$ is allowed to vary, the fit tends to optimize the atomic peak position for systems with predominantly localized two-hole states, whereas the band like intensity and shape has no further degree of freedom. Figure 4 shows a series of CS fits of the LCO data with the calculated in-plane oxygen PDOS for various fixed $U_{pp}$ including the reference fit from Fig. 3a. In order to achieve acceptable fit results, a free energy shift parameter was introduced (except for the reference fit). The energy shift of the simulated spectrum might be motivated by an uncertainty in energy referencing. Even with a free energy shift, the band like part is not well reproduced by the fit for $U_{pp}$ ($^1D$) = 4 eV ($\Delta E = 2.1$ eV), $U_{pp}$ ($^1D$) = 5 eV ($\Delta E = 1.3$ eV) or $U_{pp}$ ($^1D$) = 8 eV ($\Delta E = -1.1$ eV). Instead, the entire spectrum is correctly reproduced for $U_{pp}$ ($^1D$) = 6.3 eV ($\Delta E = 0$ eV), which suggests a model related uncertainty in the order of a few hundred meV. This conclusion requires an accurate experimental determination of the band-like part of the two-hole spectrum, which is enabled by the complete removal of the background from uncorrelated electrons by the APECS technique.

In summary, we have investigated on-site Coulomb repulsion of two-hole states in O $2p$ orbitals and the covalency of La$_2$CuO$_4$ in

comparison to CuO. With time-of-flight based APECS, the two-hole spectra were measured site-selective, which allowed us to separate the lattice oxygen from under-coordinated oxygen states at the surface and also to remove other background contributions. In the three dimensional CuO, we observe substantial delocalization pathways of the two-hole state despite $U_{pp}$ being as high as 6.8 eV. The delocalization is strongly quenched in La$_2$CuO$_4$ and the atomic multiplet is well resolved, suggesting a much decreased hybridization strength as compared to CuO. We find $U_{pp}$ = 6.3 eV for oxygen in the Cu-O planes and for apical oxygen an upper limit of $U_{pp}$ = 9.2 eV. Our findings of on-site Coulomb repulsion by APECS are relevant for advanced electronic structure models of strongly correlated solids and may contribute, e.g., to the understanding of the pairing mechanism in high temperature superconductivity.

## Methods
### Experimental station and measurement details
The APECS experiments were done at the COESCA station for electron-electron coincidence spectroscopy at the UE52-PGM beamline at the BESSY II electron storage ring[45]. The X-ray polarization was linear horizontal, the beam spot size at the sample 300 x 200 $\mu$m (h. x v.) and the photon flux from the utilized PPRE bunch[73] about $10^9$ ph/s. The two spectrometers recording the Auger electrons (AE) and photoelectrons (PE) are angle-resolving time-of-flight spectrometers (ARTOF 2 by SCIENTA) with a wide angle lens upgrade[74,75]. The photon beam, the sample surface normal, and the central axis of both spectrometers lie in the same horizontal plane. The angle between the sample normal and the photon beam is 10°, between the sample normal and the spectrometer recording the PE it is 64° and between the sample normal and the spectrometer recording the AE it is 44°. The O1$s$/ O $KVV$ APECS mesurement on CuO was done with 720 eV and for LCO with 770 eV photon energy. Both spectrometers were operated with 56° full cone acceptance and a simultaneously measured energy window of 4% of the center energy. For CuO, the center energy of the spectrometer recording photoelectrons was set to 185 eV (energy range 181.35-188.65 eV) and for Auger electrons to 509 eV (energy range 498.9-519.1 eV) and for LCO, the center energy of the spectrometer recording photoelectrons was set to 234.5 eV (energy range 229.85-239.05 eV) and for Auger electrons to 508 eV (energy range 498-518 eV). The experimental accuracy of the energies is governed by the accuracy of the time of flight determination of the Auger ARTOF. Residual uncertainty

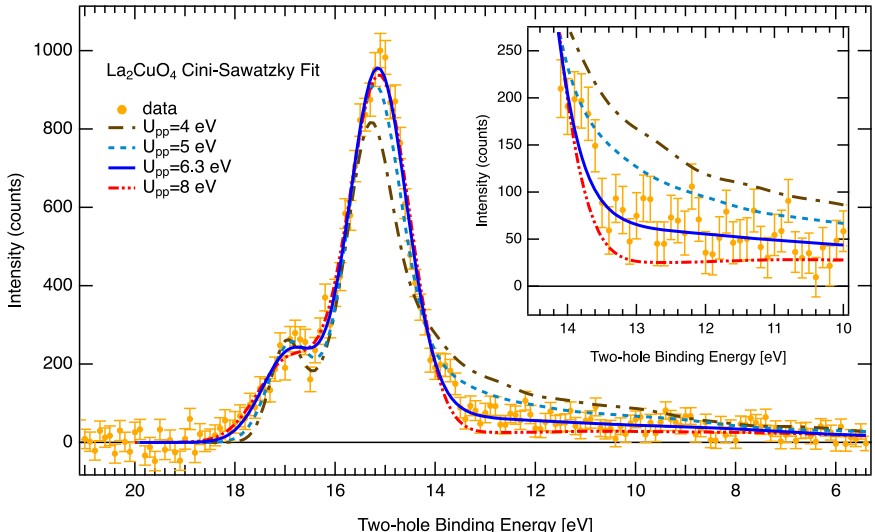

**Fig. 4 | Robustness of Cini-Sawatzky simulations in La$_2$CuO$_4$.** Variation of $U_{pp}$ and energy shift ($\Delta E$) of the PDOS leads to an optimum fit result (blue line) for $U_{pp}$ = 6.3 eV and $\Delta E$ = 0 eV. The inset zooms into the band-like final state region.

Vertical error bars are standard deviation, see Methods section. Source data are provided as a Source Data file.

of the time zero determination and alignment of the spectrometers leads to an overall uncertainty of 100 meV for the AE kinetic energy and two-hole binding energy. The uncertainty of the PE kinetic energy is about 20 meV.

## Sample fabrication and preparation

The La$_2$CuO$_4$ (LCO) film was grown by pulsed laser deposition (KrF excimer laser, $\lambda$ = 248 nm) on LaSrAlO$_4$ (001) (LSAO) substrate. The substrate holder was put at a distance of 2.5 cm from the LCO target, which was prepared by standard solid-state reaction method. During the growth, the substrate temperature was around 700 °C and the oxygen pressure around 0.8 mbar. After the growth, the film was post-annealed in vacuum at 250 °C for 30 minutes, in order to remove the presence of inadvertent excess doping oxygens, intercalated in the structure during the growth[76]. The thickness of the film is about 40 nm. The sample was transported in an evacuated bag and loaded directly into the UHV chamber. The very high sample quality was confirmed by XRD measurements, which show narrow rocking curve (FWHM = 0.15 degrees) and finite size oscillations (Supplementary Figs. 5 and 6). Moreover, survey XPS showed La, Cu, and O atoms, and some carbon presumably from surface adsorbates (Supplementary Fig. 9). In Cu 2$p$ XPS, we observe a prominent 9 eV satellite confirming copper(II)[32].

The CuO sample was prepared in the following way: First, a Cu (110) single crystal (MaTeck GmbH) was cleaned in vacuum by cycles of Argon sputtering and annealing at 600 °C. Then the sample was oxidized ex-situ on a heating plate at 300 °C for 20 min. A small contamination at the surface (mostly carbon and water) was removed by mild argon sputtering. Thereafter, partial reduction of Cu (+2) was observed, which was removed by annealing in-situ in 1 mbar oxygen for 20 min at 300 °C[32]. A clean CuO surface was confirmed by survey XPS and by observing the 9 eV satellite in Cu 2$p$ XPS and the monoclinic crystal structure was confirmed with XRD (Supplementary Figs. 7, 8 and 9).

## Data acquisition and processing

In our APECS data acquisition scheme, the flight time and detector hit position are stored together with a timestamp of the electronic trigger (synchronized to the photon pulses) for each photoelectron and Auger electron event that is registered by one of the two detectors. The resulting lists of electron events are permanently stored such that data reduction, integration and the creation of (multidimensional) histograms/ spectra is done in post-processing without losing any information stored in the primary data. Electron flight time and detector hit position are converted to kinetic energy and emission angle by non-linear transformations[45,75]. Two different electron-optical lens modes are used in the measurements. One with $\pm$ 28° full cone angular acceptance and a simultaneously measured energy window of 4% of the chosen centre energy ("Ang56_4pc") and a mode with $\pm$ 25° full cone angular acceptance and an energy window of 7% ("Ang50_7pc"). During data processing, the maximum allowed angle in the ("Ang56_4pc") mode was always restricted to $\pm$ 26° in order to exclude small energy broadening effects observed for $\theta > 26$°. The two dimensional coincidence maps are obtained by integrating over all emission angles (within the aforementioned restrictions).

Despite true coincidences, which are AE/PE pairs that originate from a single ionization event and therefore necessarily from the same photon pulse, there are also accidental (random) coincidence events detected where the two electrons originate from the same photon pulse but different ionization events. Since it is impossible to fully discriminate against accidental coincidences, the acquired total coincidence map consists always of true and accidental coincidences. With our acquisition scheme, we can accurately determine the portion of accidental coincidences within the total coincidence map and obtain the true coincidence map by subtracting a second map that contains only accidental coincidences from the total coincidence map. All

coincidence spectra (maps) presented in the manuscript are true coincidence spectra (maps), i.e. after removing the contribution from accidental coincidences. Error bars $\Delta S_{\text{true}}$ of the true coincidence spectra $S_{\text{true}}(E)$ are calculated as Poisson standard deviations of the corresponding total $S_{\text{total}}(E)$ and accidental coincidence spectra $S_{\text{acc}}(E)$ and error propagation, see Leitner et al.[45] for further details: $S_{\text{true}}(E) = S_{\text{total}}(E) - S_{\text{acc}}(E)$; $\Delta S_{\text{true}} = \sqrt{\Delta S_{\text{total}}^2 + \Delta S_{\text{acc}}^2}$ Supplementary Table 1 lists experimental settings and parameters of the APECS measurements. The main difference between the two measurements of LCO is the lens mode of the Auger electron spectrometer.

## Data availability

Source data are provided with this paper.

## Code availability

The code used for analysis and modelling in this work is available from the corresponding author upon request.

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

## Acknowledgements

The authors thank the technical staff of BESSY 2 for assistance during beamtime. The accelerator group of BESSY II is acknowledged for providing dedicated PPRE modes. D.D.C. acknowledges support by the project PRIN2020 "QT-FLUO" ID 20207ZXT4Z of the Ministry for University and Research (MUR) of Italy. K.S. and A.F. acknowledge funding from the Humboldt Research Fellowship for postdoctoral researchers programme IRL 1223611 HFST-P.

## Author contributions

D.K. and A.F. conceived the project and wrote the manuscript. D.K. and S.S. conducted the APECS measurements and analyzed the data. D.D.C and A.T. manufactured and D.D.C, A.T and K.S. characterized the $La_2CuO_4$ sample. A.F., A.L., A.T., D.D.C, D.K., F.O.L.J, K.S., N.M., S.S. discussed the results and commented on the manuscript.

## Funding

## Competing interests

The authors declare no competing interests.
