## [Transparent Peer Review file · Nature Communications]

Direct Observation of the On-site Oxygen 2p Two-Hole Coulomb Energy in La_2CuO_4

Corresponding Author: Dr Danilo Kühn

Version 0:

Reviewer comments:

Reviewer #1

(Remarks to the Author)
Key results

The paper presents a set of high quality experimental data and clearly delivers the message that Auger-Photoelectron coincidence spectroscopy (APECS) can be applied to the analysis of complex systems, such as superconductors, by highlighting electron correlation characteristics that are precluded by conventional electron spectroscopy. In particular, it shows that APECS allows not only to determine the Auger line shape with unprecedented detail, thus measuring the correlation energy U , but also to discriminate spectra characteristic of the oxygen bound in the volume of the sample from those coordinated on the surface. This is impossible with conventional Auger spectroscopy.

Validity

Both in the introduction and in the conclusions, the manuscript identifies as a major result the fact that knowledge of the correlation energy of the Auger two hole final state helps to understand the mechanism of formation of the Cooper pairs. In the introduction: "Our findings are relevant for advanced electronic structure models to understand the role of electron correlation for the pairing in cuprate high temperature superconductors and a variety of strongly correlated oxide compounds." and in the final sentence: "Our findings of on-site Coulomb repulsion by APECS are relevant for advanced electronic structure models of strongly correlated solids and may contribute, e.g., to the understanding of the pairing mechanism in high temperature superconductivity."

The ability of APECS to measure with unique discrimination on-site correlation energy of hole pairs has already been amply demonstrated by previous papers [Sbroscia, et al. 'Electron Correlation Effects in the Exchange Coupling at the Fe/CoO/Ag (001) Ferro-/Antiferro-Magnetic Interface'. JOURNAL OF MAGNETISM AND MAGNETIC MATERIALS 529 (1 July 2021). <https://doi.org/10.1016/j.jmmm.2021.167872>, and references therein quoted], this work at no point discusses connection of the measured correlation energy with the Cooper pairs formation mechanism.

Indeed, according to reference (5) of the manuscript, the electronic structure of copper oxides is influenced by the Coulomb repulsion ("the macroscopic properties of the copper oxides are profoundly influenced by strong electron-electron correlations (i.e., large Coulomb repulsion U). Naively, this is not expected to favor the emergence of superconductivity, for which electrons must be bound together to form Cooper pairs.") but the manuscript, in the present form, does not demonstrate that a strong U is connected with formation of Cooper pairs. It is exactly the latter information that could have qualified the paper for publication on a high impact journal of general interest such as Nature Communications, unfortunately it is missing.

Data and methodology

The authors rightly resort to the Cini-Sawatzky method to extract from the Auger line shape the value of the hole-hole correlation energy. Unfortunately, the implementation of the method is not correct. The major criticism is that in several cases, besides using the U value as a fit parameter of the measurement, the single particle DOS has been also arbitrarily shifted in energy, and the final feature has been arbitrarily broadened.

As a matter of fact, the main effect of the CS theory is to produce an energy shift and a line narrowing to the SCODS; to further apply an energy shift and a line broadening, correspond not using the CS theory at all, but to arbitrarily fit whatever line shape. This invalidates the whole procedure.

This criticism applies particularly to results reported in figures 3c, 4, 7. It is completely arbitrary to increase the width of the CuO

multiplet components by more than a factor 2 when the two hole binding energies are similar to the LCO ones. Moreover, in the literature broader Auger lineshapes have been attributed to disorder and inhomogeneity of the sample, i.e. different local potentials at different sites (Arena, et al. 'Line Shape of the Ag M 4,5 VV Auger Spectra Measured by Auger-Photoelectron Coincidence Spectroscopy'. *Physical Review B* 63, no. 15 (23 March 2001): 155102.

<https://doi.org/10.1103/PhysRevB.63.155102>. ; Arena, et al 'Giant Coster-Kronig Transitions and Intrinsic Line Shapes of the Anomalous Pd M45VV Auger Spectrum of Pd/Ag(100) Dilute Surface Alloys'. *PHYSICAL REVIEW LETTERS* 91, no. 17 (24 October 2003). <https://doi.org/10.1103/PhysRevLett.91.176403>. ; Ohno, 'Many-Electron Effect in the PdM45-VV APECS Spectra of Pd/Ag(100) Dilute Surface Alloys'. *JOURNAL OF ELECTRON SPECTROSCOPY AND RELATED PHENOMENA* 154, no. 3 (February 2007): 96–100. <https://doi.org/10.1016/j.elspec.2006.10.012>.). The authors should demonstrate that similar disorder effects are not responsible for the line broadening observed in CuO and LCO surface oxygen.

In figures 3a and 3b two different SCODS produces exactly the same correlated features. In literature, whenever the CS theory has been applied, this never happened. Moreover, the narrower SCDOS (apical oxygen) with the higher U (9.2 eV) is expected to produce (CS theory) a narrower correlated feature with respect to the wider SCDOS (in-plane oxygen) with the lower U; therefore, the figures show a behaviour which is in evident contrast with the CS theory. Vice versa, an identical starting SCDOS produces different correlated features for different U values, as is always the case for the atomic multiplets (Butterfield, et al.. 'Pd M 45 VV Auger Spectrum Determined by Auger-Photoelectron Coincidence Spectroscopy: Intrinsic Line Shape and Coster-Kronig Transitions'. *Physical Review B* 66, no. 11 (27 September 2002): 115115.

<https://doi.org/10.1103/PhysRevB.66.115115>.). In summary, different single particle DOS lead to different correlated features (Gotter, et al.. 'Spin-Dependent On-Site Electron Correlations and Localization in Itinerant Ferromagnets'. *PHYSICAL REVIEW LETTERS* 109, no. 12 (17 September 2012). <https://doi.org/10.1103/PhysRevLett.109.126401>.) .

All of these observations imply a low degree of discrimination in the initial hypotheses to be associated with the Auger lineshape analysis procedure applied in this work.

Notice also a typo in table 1: I(1S)/I(1D) should read I(1D)/I(1S).

Suggested improvements

Discussing in depth, if and how, the measurement of the correlation energy between two holes can help to understand the mechanism of Cooper pair formation would make the work worthy of being considered for publication on Nature Communication. Otherwise, once the other criticisms have been resolved, the work could be published in a specialised journal.

Clarity and context

Clarity and accessibility of the text is good; the results have been provided with sufficient context but consideration of previous work is barely sufficient.

References

For publications in a journal of this level it is advisable that authors do not indulge in self-citations and give credit to the works which originally reported and discussed the topics they are referring to.

Reviewer #2

(Remarks to the Author)

In this work, the authors Kuhn and others use Auger photoelectron coincidence spectroscopy (APECS) to study the onsite repulsion of oxygen 2p electrons within the CuO₂ planes in the cuprate La₂CuO₄ (LCO). Note that this method has been previously used for LCO, as described in Rietveld, G. et al. (*Physica C: Superconductivity* 185-189, 829–830). However that previous work was not conclusive. The novelty of the data provided in this preprint is that the authors are able to remove the contribution of the surface splitting that ostensibly prevented the authors in the earlier study from achieving a reasonable fit. After performing the same analysis on CuO, the authors provide limits to the Coulomb repulsion in both systems. This is a meaningful result in this subfield, and publication in Nature Communications could be suitable after the following considerations:

- 1) The article's conclusion hinges on the critical inference that the Auger line at a photoelectron energy of 532 eV in Figures 1 and 2 is the surface contribution. It would be beneficial to include a supplementary plot as shown in Figure 3 of *J. Phys. Chem. Lett.* 15, 8161 (2024) for reinforcing their main point. I would expect a strong peak at an Auger kinetic energy of 514 eV and a weaker, broader peak at 510 eV without filtering by coincidences.
- 2) In the discussion regarding Figure 4, which includes Cini-Sawatzky fits to LCO assuming different parameters for U_{pp}, the authors indicate that a free energy shift is included, which could be motivated by an uncertainty in the absolute energy of the computed partial density of states. Does this same reasoning apply to the shift that is applied for the fit for CuO referenced in Figure 3c? If so, the authors should consider moving this text earlier as the energy shift for the CuO fit is introduced without justification.

Reviewer #3

(Remarks to the Author)

Kühn et al. report Auger photoelectron coincidence spectroscopy of the O1s core level ionization and its subsequent KVV

Auger decay in the Cu(II)O systems La₂CuO₄ and (presumably) tenorite CuO₂. Fitting a Cini-Sawatzky-model to their two-hole spectra, the authors determine upper estimates of the on-site Coulomb repulsion energies for both the fully coordinated lattice- and undercoordinated surface oxygen species. I find the data and their analysis convincing, the paper well written and the conclusions to meet the overall requirements of Nature Communications. However, before publication, the authors must show evidence of a rigorous characterization of their samples to ensure reproducibility in the future:

1) Concerning the PLD grown LCO films, the authors should provide extensive RHEED, XRD and if possible, TEM data to provide evidence for their structural order, thickness, strain, defects, homogeneity, etc.

2) Same applies to the CuO data. The authors should provide extensive structural data on their films grown on Cu. Does CuO really nucleate in the conventional tenorite or in a higher symmetry tetragonal phase. What is the typical grain size and structural coherence?

3) Further, the authors should provide the XPS survey scans as well as an XPS close-up of the Cu 2p core level peaks to confirm their expected behavior.

Version 1:

Reviewer comments:

Reviewer #1

(Remarks to the Author)

The paper furtherly demonstrates that Auger-Photoelectron Coincidence Spectroscopy (APECS) is a powerful tool for analyzing systems as complex as superconductors. It highlights how APECS reveals electron correlation properties, measures correlation energy (U) with unmatched precision, and distinguishes bulk oxygen spectra from surface-coordinated ones—capabilities beyond conventional electron and Auger spectroscopy.

The revised manuscript clearly discusses plausible connections between the main experimental output U_{pp} and Cooper pairs formation mechanisms, thus qualifying the paper for publication on a high impact journal of general interest such as Nature Communications.

The additional information provided in the revised version of the manuscript addresses the concerns previously raised regarding the methods used in the analysis of the experimental data. Considering all of this, the work is now eligible for publication.

Reviewer #2

(Remarks to the Author)

I have read the authors' response to my comments, as well as the responses to the other reviewers. The authors have addressed all of my concerns in a satisfactory manner. In particular, they have provided convincing evidence that the 532 eV feature indeed originates from surface oxygen species by including the comparison between surface-sensitive coincidence measurements and bulk-sensitive non-coincidence spectra in the SI. This directly supports their key assignment and resolves my main point. Additionally, the clarification regarding the energy shift applied in the CuO fits is appropriate and places the analysis on solid footing.

Having considered these revisions together with the other reviewers' reports and the corresponding author responses, I find the manuscript now suitable for publication in Nature Communications. The work represents a significant advance in applying Auger photoelectron coincidence spectroscopy to cuprates and related oxides, providing new and meaningful insight into the Coulomb repulsion of O 2p states.

Reviewer #3

(Remarks to the Author)

Kühn et al. have satisfactorily addressed all of my earlier concerns. However, given that Referee 2 — and particularly Referee 1 — possess considerably more expertise in APECS, spectral interpretation, and the broader context of correlated cuprates, I defer to Referee 1's assessment regarding the manuscript's suitability. I therefore align with their recommendation: either the work should meet the standards of Nature Communications, or it would be better suited for a more specialized journal.

Dear Editor,

we want to thank all three reviewers for reading and commenting on our manuscript. We have carefully addressed all of the comments and we hope we could answer all concerns in the revised manuscript. We think this has helped to improve our manuscript substantially. The answers to the reviewers are given in this document in blue color. The changes in the manuscript are marked in red color.

REVIEWER COMMENTS

Reviewer #1 (Remarks to the Author):

Key results

The paper presents a set of high quality experimental data and clearly delivers the message that Auger-Photoelectron coincidence spectroscopy (APECS) can be applied to the analysis of complex systems, such as superconductors, by highlighting electron correlation characteristics that are precluded by conventional electron spectroscopy. In particular, it shows that APECS allows not only to determine the Auger line shape with unprecedented detail, thus measuring the correlation energy U , but also to discriminate spectra characteristic of the oxygen bound in the volume of the sample from those coordinated on the surface. This is impossible with conventional Auger spectroscopy.

Suggested improvements

Discussing in depth, if and how, the measurement of the correlation energy between two holes can help to understand the mechanism of Cooper pair formation would make the work worthy of being considered for publication on **Nature** Communication. Otherwise, once the other criticisms have been resolved, the work could be published in a specialised journal.

We thank the reviewer for the suggested improvements.

First, we have now deepened the discussion on the importance of our work for the field of correlated many particle states in superconductors:

The superconducting phase in LCO and many other cuprate superconductors is established by hole-doping into oxygen 2p levels in the Cu-O planes.

These holes provide charge carriers in the otherwise antiferromagnetic insulator phase of the undoped parental compound and are responsible for Cooper pair formation at low temperature whose details, however, remain under debate.

In our manuscript, we report on a strongly reduced on-site Coulomb energy of two-hole states in O 2p orbitals in the superconducting Cu-O planes as compared to the La-O planes, which we attribute to enhanced screening of the Coulomb repulsion in the Cu-O planes.

Furthermore, we observe a surprisingly strong localization of these two-hole states accompanied by strong quenching of band-like (delocalized) final states as compared to three-dimensional CuO. These important findings evidence a confinement of oxygen two-hole states to the Cu-O planes due to the favorable energetics.

The anisotropic two-particle properties differ substantially from single particle properties, where oxygen 2p PDOS DFT calculations show similar hybridization strength of oxygen in the Cu-O planes of layered LCO and of oxygen in three-dimensional CuO system.

Additionally, a quantitative knowledge of Coulomb repulsion energies of two-hole states at the relevant oxygen sites is important for theoretical models, e.g. to link electronic parameters with superexchange (Chainani, A. Physical Review B 2023, 107, 195152).

In conclusion, although on-site Coulomb repulsion measurements do not directly provide insight into the attractive force enabling Cooper pair formation, they provide important insights into the dimensionality and localization of two particle states at selected sites.

We have added the following paragraphs to the manuscript:

„Our experimental determination of on-site Coulomb energies in O 2p orbitals of LCO and CuO with Cini-Sawatzky modelling has important implications for correlated two particle states in cuprates: The on-site O 2p Coulomb energy is significantly reduced in the Cu-O planes (6.3 eV) as compared to the La-O planes (9.2 eV) probably due to enhanced screening in the Cu-O planes. This might confine the two-hole states to the Cu-O planes in the LCO material due to the favorable energetics. Furthermore, the two-hole states in LCO are strongly localized as revealed by the strong quenching of delocalized (band-like) final states in the spectra, in comparison to three-dimensional CuO. These anisotropic two-particle properties in LCO differ substantially from single particle properties, where oxygen 2p PDOS DFT calculations show similar hybridization strength of oxygen in the Cu-O planes of layered LCO and of oxygen in three-dimensional CuO. We thus think that the observed properties of the Cu-O planes, with regard to two hole states, relate to the preference of pairing in cuprate high temperature superconductors in the Cu-O planes alike. A quantitative knowledge of Coulomb energies of two-hole states at oxygen sites is thus relevant for theoretical models, e.g. to link electronic parameters with superexchange [57].“

Secondly, we now discuss our implementation of the Cini-Sawatzky model in more detail. In short, we have addressed the necessity of additional broadening of the CS shapes, which have zero intrinsic width in the strong correlation regime. This broadening is motivated by phonon broadening, potential variations in the surface region, lifetime broadening and other effects. We agree with the reviewer that the extra broadening was not sufficiently motivated in the manuscript. We now put the broadening of the multiplet states into relation to the O1s photoemission peak width to explain the underlying mechanism and the differences between the broadenings observed for LCO and CuO.

We want to note that all applied broadenings to all LCO and CuO spectra are much smaller than the widths of the respective non-interacting PDOS's.

We discuss why CS modelling of Auger spectra in CuO is difficult, probably due to strong hybridization between the valence states, and why an energy shift might be justified due to uncertainties in energy alignment in this case, whereas no energy shift is needed in the LCO spectra.

We give further justification for our assignment of the main features in the LCO spectrum being a superposition of in-plane oxygen and apical oxygen. This unambiguous assignment is possible due to the high statistical significance of our data excluding hidden features at other binding energies.

We have added additional references to substantiate the discussion of the broadening mechanisms in the two-hole spectra and the integration of our work into context of previous APECS work on strongly correlated systems.

See a detailed reply to the comments in the data and methodology section down below.

We have added the following paragraphs to the manuscript:

„...A similar approach was previously used to model APECS spectra from transition metal surfaces [61–63] and magnetic thin films [64,65] in order to probe electronic correlations.“

„...This broadening accounts for effects that are not captured by the Cini-Sawatzky model: After removing experimental contributions, the line width of the multiplet states is similar to the O 1s width (1.1 eV) of lattice oxygen, which suggests a related broadening mechanism present in O 1s and O KVV peaks. Therefore, we favor to attribute the broadening to phonon excitations or small variations of the local potential, which should both affect the PES and AES similarly.[66] Also a small life time broadening and dispersion of the two-hole state might contribute.“

„...The fitted Gaussian broadening of 4.3 eV is substantially bigger than in LCO and than the O 1s line width of CuO lattice oxygen. This suggests effects beyond phonon broadening or variation of the local potential, as, e.g., coupling to the copper ions.

Clarity and context

Clarity and accessibility of the text is good; the results have been provided with sufficient context but consideration of previous work is barely sufficient.

References

For publications in a journal of this level it is advisable that authors do not indulge in self-citations and give credit to the works which originally reported and discussed the topics they are referring to.

Validity

Both in the introduction and in the conclusions, the manuscript identify as a major result the fact that knowledge of the correlation energy of the Auger two hole final state helps to understand the mechanism of formation of the Cooper pairs. In the introduction: “Our findings are relevant for advanced electronic structure models to understand the role of electron correlation for the pairing in cuprate high temperature superconductors and a variety of strongly correlated oxide compounds.” and in the final sentence: “Our findings of on-site Coulomb repulsion by APECS are relevant for advanced electronic structure models of strongly correlated solids and may contribute, e.g., to the understanding of the pairing mechanism in high temperature superconductivity.”

The ability of APECS to measure with unique discrimination on-site correlation energy of hole pairs has already been amply demonstrated by previous papers [Sbroscia, et al. ‘Electron Correlation Effects in the Exchange Coupling at the Fe/CoO/Ag (001) Ferro-/Antiferro-Magnetic Interface’. JOURNAL OF MAGNETISM AND MAGNETIC MATERIALS 529 (1 July 2021). <https://doi.org/10.1016/j.jmmm.2021.167872>, and references therein quoted], this work at no point discusses connection of the measured correlation energy with the Cooper pairs formation mechanism. Indeed, according to reference (5) of the manuscript, the electronic structure of copper oxides is influenced by the Coulomb repulsion (“the macroscopic properties of the copper oxides are profoundly influenced by strong electron-electron correlations (i.e., large Coulomb repulsion U). Naively, this is not expected to favor the emergence of superconductivity, for which electrons must be bound together to form Cooper pairs.”) but the manuscript, in the present form, do not demonstrate that a strong U is connected with formation of Cooper pairs. It is exactly the latter information that could have qualified the paper for publication on a high impact journal of general interest such as Nature Communications, unfortunately it is missing.

Data and methodology

The authors rightly resort to the Cini-Sawatzky method to extract from the Auger line shape the value of the hole-hole correlation energy. Unfortunately, the implementation of the method is not correct. The major criticism is that in several cases, besides using the U value as a fit parameter of the measurement, the single particle DOS has been also arbitrarily shifted in energy, and the final feature has been arbitrarily broadened. As a matter of fact, the main effect of the CS theory is to produce an energy shift and a line narrowing to the SCODS; to further apply an energy shift and a line broadening, correspond not using the CS theory at all, but to arbitrarily fit whatever line shape. This invalidate the whole procedure.

This criticism applies particularly to results reported figures 3c, 4, 7. It is completely arbitrary to increase the width of the CuO multiplet components by more than a factor 2 when the two hole binding energies are similar to the LCO ones. Moreover, in the literature broader Auger lineshapes have been attributed to disorder and inhomogeneity of the sample, i.e. different local potentials at different sites (Arena, et al. ‘Line Shape of the Ag M 4,5 VV Auger Spectra Measured by Auger-Photoelectron Coincidence Spectroscopy’. Physical Review B 63, no. 15 (23 March 2001): 155102. <https://doi.org/10.1103/PhysRevB.63.155102> ; Arena, et al ‘Giant Coster-Kronig Transitions and Intrinsic Line Shapes of the Anomalous Pd M45VV Auger Spectrum of Pd/Ag(100) Dilute Surface Alloys’. PHYSICAL REVIEW LETTERS 91, no. 17 (24 October 2003). ; Ohno, <https://doi.org/10.1103/PhysRevLett.91.176403>. ‘Many-Electron Effect in the PdM45-VV APECS Spectra of Pd/Ag(100) Dilute Surface Alloys’. JOURNAL OF ELECTRON SPECTROSCOPY AND RELATED PHENOMENA 154, no. 3 (February 2007): 96–100. <https://doi.org/10.1016/j.elspec.2006.10.012>.) The authors should demonstrate that similar disorder effects are not responsible for the line broadening observed in CuO and LCO surface oxygen.

In figures 3a and 3b two different SCODS produces exactly the same correlated features. In literature, whenever the CS theory has been applied, this never happened. Moreover, the narrower SCDOS (apical oxygen) with the higher U (9.2 eV) is expected to produce (CS theory) a narrower correlated feature with respect to the wider SCDOS (in-plane oxygen) with the lower U ; therefore, the figures show a behaviour which is in evident contrast with the CS theory. Vice versa, an identical starting SCDOS produces different correlated features for different U values, as is always the case for the atomic multiplets (Butterfield, et al.. ‘Pd M 45 VV Auger Spectrum Determined by Auger-Photoelectron Coincidence Spectroscopy: Intrinsic Line Shape and Coster-Kronig Transitions’. *Physical Review B* 66, no. 11 (27 September 2002): 115115. <https://doi.org/10.1103/PhysRevB.66.115115>). In summary, different single particle DOS lead to different correlated features (Gotter, et al.. ‘Spin-Dependent On-Site Electron Correlations and Localization in Itinerant Ferromagnets’. *PHYSICAL REVIEW LETTERS* 109, no. 12 (17 September 2012). <https://doi.org/10.1103/PhysRevLett.109.126401>).

All of these observations imply a low degree of discrimination in the initial hypotheses to be associated with the Auger lineshape analysis procedure applied in this work.

Notice also a typo in table 1: I(1S)/I(1D) should read I(1D)/I(1S).

Reply data and methodology:

The CS method allows to simulate interacting two particle DOS from non-interacting single particle DOS, as described by *Cini, M. Solid State Commun. 1977, 24, 681—684.* ; *Sawatzky, G. A. Phys. Rev. Let. 1977, 39, 504—507.*

In our manuscript, we use the numerical expression by "*Cini, M. Solid State Commun. 1977, 24, 681—684.*", which was previously mostly used for metallic and alloy systems, to simulate and fit oxygen Auger spectra of LCO and CuO.

We tested our implementation of the numerical routines carefully and could fully reproduce the simulations of Figure 1 in "*Cini, M. Solid State Commun. 1977, 24, 681—684.*"

We are delighted that the reviewer agrees with us that the CS model is a suitable framework to obtain the on-site Coulomb repulsion U in oxygen two-hole states.

However, to our best knowledge, it was previously not applied to O KVV spectra of oxides in literature, hence one goal of the manuscript is indeed to investigate how applicable this model is for oxygen two-hole states in strongly correlated materials.

We want to point out that Cini's formula was derived for a single band system and it is not expected to reproduce Auger spectra of transition metal oxides with hybridization between the metal and oxygen bands in all details even though the hybridization is taken into account to some extent in the PDOS calculations.

Energy shifts and broadenings:

In a fit based on Cini's formula, for a given single particle DOS, U is the only free parameter.

It was discussed in „*Sawatzky, G. A.; LenseLink, A. Phys. Rev. B 1980, 21, 1790—1796*” that, even for relatively small Coulomb repulsion U , the interacting two-hole DOS is already strongly peaked and, as a consequence, the fit routine will always try to match this peak to the peak maximum of the experimental data, no matter how well (or poor) the rest of the spectral range is matched.

In the case of O 2p two-hole states, the experimental spectra reveal a rather high Coulomb energy for LCO, which can be roughly measured as the energy distance between the maximum of the calculated self-convoluted DOS and the maximum of the experimental two-hole spectrum (about 7 eV). The so obtained U is overestimated by about 10-20% („*Sawatzky, G. A.; LenseLink, A. Phys. Rev. B 1980, 21, 1790—1796*“).

In this regime ($U \approx W$), the bound state becomes very narrow and can become a delta peak split off from the band-like part ("*Cini, M. Solid State Commun. 1977, 24, 681—684.*" ; „*Sawatzky, G. A.; LenseLink, A. Phys. Rev. B 1980, 21, 1790—1796*“).

Therefore, we think it is necessary and justified to introduce a broadening of the simulated CS spectra, to account for several effects, i.e. experimental resolution, phonon broadening, dispersion of the two-hole state, coupling to the copper ions, small variations of the local potential in the

surface region („Greczynski, G. et al. *Nature Reviews Methods Primers* 2023, 3, 40“) and life time broadening.

After deconvoluting the experimental AES resolution (0.5 eV), the broadenings of in-plane and apical oxygen in Fig. 3a and b (1.3 eV, 1.5 eV) are comparable to the 1.1 eV line width of the O 1s photoemission peak

This suggests that the reasoning of the line width of the O 1s lattice oxygen peak and the O KVV bound state are of similar nature.

Phonon broadening or a small variation of the local potential in the surface region of the poorly conducting oxides could most stringently explain the similar broadening in PES and AES.

Such broadening is not captured by the CS formula and needs to be incorporated explicitly.

However, the extra broadening does not render the CS fit arbitrary, because an important part of the CS model is to estimate the shape of the band-like part and the bound to band-like intensity ratio.

These are little affected by a Gaussian broadening as long as the broadening is not very large.

We find an excellent agreement of the CS fit for LCO in-plane oxygen for the full spectral range without any shifting of the DOS.

Since energy alignment between electron spectra of oxide insulators („Greczynski, G. et al. *Nature Reviews Methods Primers* 2023, 3, 40“) and calculated DOS is not trivial, in Fig. 4, we test the sensitivity of the CS fit to an intentional shift of the DOS.

We find excellent agreement between fit and experimental data with the unshifted DOS and we show that shifts about ± 1 eV already distorts the fit noticeably. This evidences that the energy alignment in LCO is correct and hence gives further credibility to the obtained U.

In-plane and apical oxygen:

The LCO two-hole spectrum clearly shows two distinct features at 514 eV and 512 eV. The intensity ratio and energy splitting makes it very plausible to assign them to the 1D and 1S peaks of the O $2p^4$ atomic multiplet.

Due to the high statistical significance of our data we can estimate an upper limit of 5% of the intensity of the „514eV/512eV doublet structure“ for a hypothetical second atomic multiplet "hidden" at different two-hole binding energy.

Since the stoichiometry of in-plane oxygen to apical oxygen is 1:1 in bulk LCO, it is very unlikely that either the in-plane or the apical oxygen are "hidden" in the spectrum at a different binding energy than the „514eV/512eV doublet structure“.

Although the intensity of both oxygen species is expected to deviate from the bulk stoichiometry, because of the very-high surface sensitivity (1-2 layers mean escape depth) of our APECS measurement in combination with a preferential La-O termination of LCO surfaces, this effect can at most modify the in-plane to apical oxygen intensity ratio by roughly a factor of three and hence a second atomic multiplet could not be fully hidden in the spectrum.

Therefore, we conclude that the atomic multiplets of in-plane and apical oxygen must show up at same two-hole binding energies (within the peak envelopes) in the „514eV/512eV doublet structure“.

Unfortunately, at present, we can not resolve the in-plane and apical oxygen multiplets and we decided to do two separate CS fits based on the respective in-plane and apical oxygen DOS's, which stringently gives the Coulomb repulsions of 6.3 eV and 9.2 eV, respectively, as presented in the manuscript.

In principle it should be possible that different uncorrelated two particle DOS's produce similar correlated atomic features at same two-hole binding energy if also U differs by the „right“ amount. In the future, we will investigate for different cuprates whether the identical two-hole binding energies of in-plane and apical oxygen are a particularity of LCO or whether this is a general behavior of the layered cuprates.

Analysis of O KVV of CuO:

The main differences between the O 2p two-hole spectra of LCO and CuO are a much broader atomic multiplet and a strongly increased band-like part in CuO. The width of the O 1s photoemission peak of CuO lattice oxygen is about 1.2 eV, comparable to the O 1s width of LCO. Hence, a broadening of 1.2 eV due to variation of local potential or phonon broadening is also expected in the O KVV of CuO. However, the much increased width of about 4 eV and the strong band-like part must be attributed to additional effects not present in LCO. We suggest two possible explanations:

First, the covalency of the O 2p orbitals might be stronger in the three dimensional CuO compared to the layered LCO, which might explain the enhanced band-like contribution in CuO.

Furthermore, the strongly broadened atomic part might be explained by stronger coupling to the Cu ions causing an unresolved splitting of the atomic multiplet in CuO.

Unfortunately, only few high resolution O KVV data of transition metal oxides are reported in literature. Comparing our data to Fig. 2 in "Fuggle, J. J. of Electron Spectroscopy and Related Phenomena 1982, 26, 111–132" it appears that the majority of O KVV spectra of oxides have rather broad atomic peaks, as in our CuO spectrum, and only the strongly ionic NaO shows a narrow peak comparable to our LCO spectrum.

Concerning the fits, it is clear that our CS fit for CuO based on unshifted LDA-DFT PDOS gives fairly poor results. Even with an extra broadening in the CS model, the band-like contribution remains much underestimated. We propose two ways to explain and resolve this discrepancy: First, the CS model might be insufficient to describe the CuO O KVV spectrum in detail because of the involved approximations, i.e. single-band system approximation and neglect of Auger matrix elements. This could lead to an underestimation of the intensity of the band-like final states.

Secondly, we also want to consider the possibility of issues with the energy referencing of the calculated oxygen PDOS and the experimental data. This is why we allow the DOS to shift in a second fit (and only in this one). The result gives a much better agreement. We want to point out that a very similar fit result can be obtained with LDA+U PDOS (SI Fig. 3).

While the use of DOS which already includes U might be questionable "Ramaker, D. E. *Critical Reviews in Solid State and Material Sciences* 17.3 (1991): 211-276", we think it is important to also report on U of the fit with shifted DOS and LDA+U DOS.

We interpret the results for U based on the CS fits with unshifted PDOS and optimally shifted PDOS as upper and lower boundaries of U. It is at present not fully understood why the energy referencing seems correct for LCO, but questionable for CuO. One possibility is that the defect states at the surface provide charge carriers more effectively in the LCO surface region than in CuO. We believe that our CS fits and presented U values are robust and technically correct. However, more sophisticated ab-initio calculations are required to understand the two-hole spectra in detail.

We thank the reviewer for pointing out the mistypo „I(1S)/I(1D)“ in Table 1.

Reviewer #2 (Remarks to the Author):

In this work, the authors Kuhn and others use Auger photoelectron coincidence spectroscopy (APECS) to study the onsite repulsion of oxygen 2p electrons within the CuO₂ planes in the cuprate La₂CuO₄ (LCO). Note that this method has been previously used for LCO, as described in Rietveld, G. et al. (*Physica C: Superconductivity* 185-189, 829–830). However that previous work was not conclusive. The novelty of the data provided in this preprint is that the authors are able to remove the contribution of the surface splitting that ostensibly prevented the authors in the earlier study from achieving a reasonable fit. After performing the same analysis on CuO, the authors provide limits to the Coulomb repulsion in both systems. This is a meaningful result in this subfield, and publication in Nature Communications could be suitable after the following considerations:

1) The article's conclusion hinges on the critical inference that the Auger line at a photoelectron energy of 532 eV in Figures 1 and 2 is the surface contribution. It would be beneficial to include a supplementary plot as shown in Figure 3 of J. Phys. Chem. Lett. 15, 8161 (2024) for reinforcing their main point. I would expect

a strong peak at an Auger kinetic energy of 514 eV and a weaker, broader peak at 510 eV without filtering by coincidences.

We thank the reviewer for this comment.

Indeed, we assign the O 1s photoemission feature at about 532 eV binding energy to surface oxygen species such as under-coordinated oxygen and adsorbed species.

While an increase in binding energy of such species with respect to well coordinated lattice oxygen is well reported for cuprates and other oxides in literature (e.g. ref. 17, 30, 32 and 46 in manuscript), we now also show direct evidence that the “532 eV species“ is accumulated at the surface.

As the reviewer suggests, one way is to compare the more surface sensitive AECS and PECS with the more bulk sensitive non-coincidence AES and PES (e.g. ref. 44,51 in manuscript).

We have included Fig. 1 in the SI, which shows these spectra for LCO and CuO.

In both O 1s PECS spectra it is clear that the component at 532 eV is strongly enhanced with respect to the component at about 529-530 eV, which directly proves that the „532 eV species“ is indeed a surface species.

In the AECS of LCO, the broad component at 510 eV kinetic energy is enhanced with respect to the component at 514 eV, making it reasonable to attribute the broad 510 eV peak to predominantly arising from the surface species.

We added following sentence to the main text:

„...The depth distribution of these features can be directly inferred from their relative intensities in the very surface sensitive coincidence measurement in comparison to the deeper probing non-coincidence measurement (see SI Fig. 1).“

2) In the discussion regarding Figure 4, which includes Cini-Sawatzky fits to LCO assuming different parameters for U_{pp} , the authors indicate that a free energy shift is included, which could be motivated by an uncertainty in the absolute energy of the computed partial density of states. Does this same reasoning apply to the shift that is applied for the fit for CuO referenced in Figure 3c? If so, the authors should consider moving this text earlier as the energy shift for the CuO fit is introduced without justification.

We thank the reviewer for this comment. We agree that a justification for the energy shift in the CS simulation of the CuO spectrum was missing. We added following sentence to this section:

„...We relate all data to the common Fermi level of a metallic sample which still allows for surface shifts in doped gaped materials.[66] For LCO, defect states provide a higher charge carrier density than for CuO.“

Reviewer #3 (Remarks to the Author):

Kühn et al. report Auger photoelectron coincidence spectroscopy of the O1s core level ionization and its subsequent KVV Auger decay in the Cu(II)O systems La₂CuO₄ and (presumably) tenorite CuO₂. Fitting a Cini-Sawatzky-model to their two-hole spectra, the authors determine upper estimates of the on-site Coulomb repulsion energies for both the fully coordinated lattice- and undercoordinated surface oxygen species. I find the data and their analysis convincing, the paper well written and the conclusions to meet the overall requirements of **Nature Communications**. However, before publication, the authors must show evidence of a rigorous characterization of their samples to ensure reproducibility in the future:

1) Concerning the PLD grown LCO films, the authors should provide extensive RHEED, XRD and if possible, TEM data to provide evidence for their structural order, thickness, strain, defects, homogeneity, etc.

We thank the reviewer for the comment. Indeed, a comprehensive characterization of the LCO film was missing in the manuscript. We now present the XRD data in the SI (see Fig. 5) and reciprocal space mapping measurements (Fig. 6), which were measured on the same film as was used for the APECS measurements. Additionally, we have done X-ray reflectivity measurements on a twin LCO film grown with the identical setup under same growing conditions. We have added the following paragraph to the supplementary:

“The structural quality of the LCO films was characterized using X-ray diffraction (XRD), reciprocal space mapping (RSM) and X-ray reflectivity (XRR). The symmetric XRD scan in Fig. 5(a) confirms single phase epitaxial growth of LCO on the SLAO(001) substrate. Well evident finite size oscillations and narrow (FWHM < 0.1°) rocking curve, shown in Figs. 5(b) and 5(c), respectively, indicate uniform film thickness, smooth interfaces and high crystalline quality, with low density of extended defects.

The reciprocal space map (RSM) in Fig. 6, measured around the SLAO (107) reflection, enables the determination of both in-plane and out-of-plane lattice parameters of the LCO film. While the film likely experiences substrate-induced strain near the interface, its thickness allows for strain relaxation. The measured lattice parameters, $a = 381 \pm 1$ pm and $c = 1312 \pm 4$ pm, indicate that the film has relaxed toward its bulk structure. XRR measurements (not shown) were done on a second LCO sample grown with the identical setup under same conditions. The XRR measurements indicate a film thickness of 56 nm and a surface roughness of approximately 2 nm.

2) Same applies to the CuO data. The authors should provide extensive structural data on their films grown on Cu. Does CuO really nucleate in the conventional tenorite or in a higher symmetry tetragonal phase. What is the typical grain size and structural coherence?

We agree with the reviewer that a stringent characterization of the CuO film was missing.

We now characterized the oxidized copper crystal with XRD and we show the data in the SI (Fig. 7 and 8).

In short, with XRD we identify a clear contribution from a monoclinic CuO overlayer with predominantly 111 orientation.

We also observe a substantial Cu₂O signal and contribution from the copper single crystal substrate with 110 orientation.

The Cu₂O (Cu +1 oxidation state) deeper in the crystal arises from various oxidation cycles within the measurement series.

Each preparation cycle has involved oxygen dosing and ion etching with subsequent annealing above 600°C in vacuum, which reduces the CuO overlayer to the thermodynamically stable Cu₂O phase at elevated temperature.

This has subsequently formed a thick Cu₂O layer.

Therefore, the CuO layer formed in the final preparation step by oxygen dosing in vacuum as measured in this work is on top of a thick Cu₂O layer with some remaining unoxidized regions deeper in the crystal.

We have added the following paragraph to the SI:

“To characterize the structure of the film grown on Cu, we performed X-ray diffraction (XRD) measurements. Symmetric XRD scans using a parallel beam configuration ($\theta = \omega$) primarily probe lattice planes that are parallel to the sample surface, making this geometry well-suited for confirming the orientation of single crystals such as the underlying Cu (110). However, this configuration is less sensitive to misoriented or polycrystalline phases.

To enhance the visibility of diffraction peaks from the oxidized overlayer, which may exhibit various orientations, we offset the incidence angle ω by a small amount (e.g., 1°). This allows detection of off-normal planes and improves sensitivity to non-epitaxial phases.

The symmetric XRD scan in Fig. 7 confirms the (110) orientation of the Cu single crystal substrate. To identify the oxide phases and assign the observed diffraction peaks, we analysed the data collected with the 1° ω -offset scan.

The measured XRD pattern, shown in Fig. 8, was compared to reference patterns of cubic Cu₂O, monoclinic CuO, and orthorhombic CuO₂. No peaks corresponding to orthorhombic CuO₂ were observed. All visible reflections can be attributed to cubic Cu₂O and monoclinic CuO.

The data indicate that the Cu₂O layer formed on the Cu(110) single crystal is polycrystalline and exhibits some degree of texturing. The overlying CuO layer is also polycrystalline; however, a pronounced (111) texture appears to dominate, likely due to the lower surface energy associated with this orientation.

Analysis of the CuO(111) reflection using the Scherrer equation yields a structural coherence length of approximately 17 ± 2 nm along the [111] direction. This value represents the average

distance over which the CuO lattice remains well-ordered and free from significant disruptions such as stacking faults, dislocations, or grain boundary misorientations. It serves as a lower bound for the actual grain size.

In the literature, CuO films formed by oxidation of Cu_2O exhibit grain sizes ranging from 20 to 120 **nm**, depending on oxidation conditions [Mahana D. et al., *Solid State Commun.* **366–367**, 115152 (2023)]. The structural coherence length estimated in our study ($\sim 17 \pm 2$ nm) is consistent with the lower end of this range, indicating the presence of well-ordered crystalline domains of comparable scale.

3) Further, the authors should provide the XPS survey scans as well as an XPS close-up of the Cu 2p core level peaks to confirm their expected behavior.

We thank the reviewer for the comment. We have now added a Figure in the SI showing the XPS survey scans of LCO and CuO, as well as Cu 2p XPS. The spectra demonstrate a very clean CuO surface and a clean LCO surface with some adsorbates at the surface indicated by a small carbon peak. The Cu 2p spectra show characteristic 3d₉ satellites, which confirm the oxidation state of copper to be +2 in both samples.